# Greater trophic diversity of soil animal communities under agricultural land use and tropical climate

Zheng Zhou[1,2] ✉, Nico Eisenhauer[3,4], Andrew D. Barnes[5], Melanie M. Pollierer[1,6], Malte Jochum[7], Ingo Grass[2,8], Yan Zhang[1], Ulrich Brose[3,9], Fujio Hyodo[10], Nicole Scheunemann[11], Olaf Schmidt[12], Yuanyuan Huang[3,4], Bernhard Klarner[1], Anton A. Goncharov[13], Alena Krause[1], Daniil Korobushkin[13], Anastasia Gorbunova[13,14], Ilya I. Lyubechanskii[15,16], Sergey M. Tsurikov[13], Julia Seeber[17,18], Michael Steinwandter[17], Vladimir A. Zryanin[19], Oksana L. Rozanova[13], Winda Ika Susanti[1], Felicity V. Crotty[20], Di Ajeng Prameswari[1], Zhipeng Li[1,21], Carol Melody[12], Zhijing Xie[22], Xue Pan[1], Donghui Wu[22], Mark Maraun[1], Katerina Sam[23,24], Alexei V. Tiunov[13,25,28], Stefan Scheu[1,26,28] & Anton Potapov[1,3,11,27,28]

Soil fauna contributes to a wide range of ecosystem functions via their trophic activities. Here we investigate how trophic diversity of soil animals varies across functional groups and major biomes. We use stable isotope analysis ($^{13}C/^{12}C$ and $^{15}N/^{14}N$ ratios) of 17,306 samples of 28 high-rank taxa from 456 sites across 19 countries to inspect the variability in trophic diversity across climate regions and land-use types. Trophic diversity of soil animal communities is higher for microbial feeders than for detritivores and predators, in agricultural ecosystems compared with woodlands (+32%) and in tropical compared with temperate climates (+40%). Higher trophic diversity is related to more diverse basal resources and longer trophic chains, which could reflect greater niche partitioning in resource-limited environments. Our findings suggest that soil animals could broaden their trophic niches under agricultural land use and possibly in response to warming, but whether such foraging flexibility may offset the loss of trophic specialists remains to be investigated.

Soils are the most biodiverse habitats on Earth contributing to about 59% of global biodiversity[1]. Approximately 90% of the carbon fixed by plants in terrestrial ecosystems enters the belowground system[2] and is processed in soil food webs by microorganisms and invertebrate decomposers, with the latter serving as prey for predators[3,4]. Soil food webs are characterized by major energy fluxes in terrestrial ecosystems and trophic interactions among an exceptionally diverse spectrum of organisms with different niches. Consequently, soil food webs are of essential importance for carbon and nitrogen cycling and thereby for

ecosystem functions and services[5]. Beyond belowground processes, trophic interactions of soil organisms extend to the biodiversity and functionality of the aboveground system, fostering feedback loops between aboveground and belowground compartments of terrestrial ecosystems[3,6,7]. It has been shown that the functional diversity of soil animals (rather than species richness alone) is closely associated with litter decomposition, carbon and nutrient cycling and plant growth[8–11]. The diversity of functions driven by soil animals in food webs is based on their trophic diversity, defined as the heterogeneity of trophic niches

(including basal resources and trophic levels) across soil animal individuals and taxa[5,8]. Trophic diversity offers insight into community structure and function beyond taxonomic diversity. Unravelling the factors influencing trophic niches and trophic diversity is therefore crucial for understanding species coexistence and ecosystem stability and functionality[12,13].

Soil animals fulfil diverse roles within soil food webs and are often classified into functional groups[14,15]. For instance, detritivores serve as primary decomposers breaking down and consuming dead plant material, thereby contributing to decomposition and other processes[16–18]. Microbivores, as secondary decomposers, influence the growth and dispersal of prokaryotes and fungi, indirectly regulating nutrient cycling by changing the activity and composition of microbial communities[19–21]. Predators play a crucial role in population regulation and maintenance of biodiversity through top-down control[2,22,23]. Understanding the trophic diversity within these functional groups and how they respond to environmental changes, such as land use and climate, is crucial for understanding the ecosystem functions they provide.

Across the globe, land-use change alters the composition of ecological communities and often leads to a decline in ecosystem functions, which are at the core of sustainable development goals[24–26]. Land-use changes the structure of and energy flux through soil food webs and also shifts trophic positions of soil animals[27–30], thereby influencing the stability and functioning of ecosystems[31]. Studies showed that land use affects the trophic diversity of various animal groups, such as fish, birds, mammals and zoobenthos[32–36]. However, it is still not clear how land use alters the trophic diversity of soil animals and whether land-use effects differ among functional groups of soil animals across climatic regions. Moreover, recent global assessments of soil animals demonstrated changes in richness and density with latitude[37–39], highlighting the climate-driven changes in soil biodiversity and functions. Existing studies mainly focus on changes in the taxonomic and morphological diversity of soil animal communities, showing that they decrease with land-use intensity[39–41], whereas little is known about changes in the trophic diversity of soil animal communities.

Trophic diversity of animal functional groups is shaped by different, non-mutually exclusive mechanisms: large trophic diversity may be due to either greater within-taxon trophic diversity (niche expansion) or greater between-taxon trophic dissimilarity (niche partitioning among taxa)[32,42]. Furthermore, large trophic diversity may be a consequence of a broad range of basal resources or a high number of trophic levels[42,43]. Environmental changes driven by land-use intensification may lead to increased generalism due to higher resource limitation, resulting in trophic homogenization and thus lower diversity via bottom-up constraints[42].

Here we compiled a dataset comprising 17,306 records on 28 high-rank taxa (broad taxonomic groups, such as Collembola, Araneae and Lumbricina) of soil animals across 456 sites (Fig. 1), leveraging published and unpublished stable isotope ($\delta^{13}$C, $\delta^{15}$N) data to investigate the differences in trophic diversity of soil animal taxa across functional groups and biomes and explore the underlying mechanisms. The $\delta^{13}$C in consumer tissue provides insight into basal resources used by consumers (from fresh plant material, to dead leaves and microbially processed soil organic matter[43]), whereas the $\delta^{15}$N reflects their trophic level[43,44]. To reflect trophic diversity, we used sample-size corrected standard stable isotope ellipse areas in the $\delta^{13}$C–$\delta^{15}$N space, which captures variation in trophic niches within a group (the range of basal resources and trophic levels across individuals/taxa)[45]. We used the isotopic distance among taxa within functional groups to reflect trophic dissimilarity[45–47]. We explored variation in trophic diversity and dissimilarity among functional groups of soil animals (detritivores, microbivores, predators and herbivores) and how these trophic traits are modulated by land use (woodlands versus agricultural ecosystems) and climatic region (temperate versus tropical). We hypothesized that (1) trophic diversity is higher for microbivores and herbivores compared with detritivores and predators because of more pronounced niche partitioning among taxa within these functional groups[15]; and (2) trophic diversity of soil animals is larger in biomes with higher taxonomic diversity, specifically in woodlands and tropical regions[37–39] compared with agricultural ecosystems[48] and temperate regions[49], respectively. In addition, we explored the mechanisms contributing to differences among functional groups and biomes, by testing whether niche expansion or partitioning explains differences in trophic diversity among functional groups and biomes and if variations in trophic diversity are related to both variations in basal resources ($\delta^{13}$C) and trophic levels ($\delta^{15}$N).

## Results and discussion
### Trophic diversity differs among functional groups

Overall, the trophic diversity and niche differentiation depend on the position of functional groups within the food web (Fig. 2a, Supplementary Fig. 1a and Supplementary Table 1). Conforming to our first hypothesis, the trophic diversity (corrected standard ellipse area, SEAc) of microbivores was 61.6% and 69.0% greater than that of detritivores and predators, respectively (Fig. 2a and Supplementary Table 1). Trophic diversity of single taxa within functional groups exhibited similar patterns to the trophic diversity of their respective overarching functional groups (Fig. 2a,b and Supplementary Table 2). Larger trophic diversity resulted from both higher variability in $\delta^{13}$C and $\delta^{15}$N values as indicators of variability in the use of basal resources and in trophic level, respectively (Fig. 2c,d, Supplementary Fig. 2b and Supplementary Table 4). Besides, both trophic diversity of individual taxa within functional groups (niche expansion) and trophic dissimilarity among taxa (niche partitioning) contributed to the overall trophic diversity of functional groups, with the influence of niche expansion being stronger (Fig. 2e,f, Supplementary Fig. 2a and Supplementary Table 5).

Microbivores exhibited larger trophic diversity compared with detritivores and predators, due to the combination of higher variability in basal resources and trophic levels (Supplementary Fig. 3a,b). Presumably, the small body size of microbivores enables them to access a wide range of microhabitats within the pore space of soils[15,50], allowing microbivores to exploit a diverse spectrum of microorganisms with distinct stable isotope compositions[43]. This may result in a wider range of trophic niches and exploitation of a larger diversity of basal resources compared with, for example, detritivores. It has been suggested that microorganisms are analogues of animals with distinct trophic levels[51]. Consequently, by feeding on different microorganisms, microbivores are likely to vary in stable isotope values. In fact, it has been documented that microbivores, such as springtails and oribatid mites, span a wide range of trophic levels[52,53], while detritivores have narrower ranges[54,55] and this is confirmed by the large variation in $\delta^{15}$N values of microbivores in our study (Supplementary Fig. 3b and Supplementary Table 4). Contrary to microbivores, detritivores are larger and more mobile and are therefore likely to integrate a wider range of food resources by foraging at larger spatial scales than microbivores[56]. Combined with the high incidence of generalist feeding in detritivores[15,42], their similar feeding habits often lead to overlapping trophic niches among individuals, resulting in more homogeneous trophic niches and reduced overall trophic diversity at functional group level (see Fig. 1c in ref. 42).

Predatory taxa had both smaller trophic diversity (Fig. 2b and Supplementary Table 2) and trophic dissimilarity (Supplementary Fig. 1a and Supplementary Table 3) compared with other functional groups, indicating that they not only exhibit lower trophic diversity within each predator taxon (lower niche expansion), but also with similar trophic niches among different predator taxa (lower niche partitioning). This similarity contributed to the overall reduced trophic diversity among predators compared with other functional groups (Fig. 2a and Supplementary Table 1). Soil predators tend to be generalists and hunt

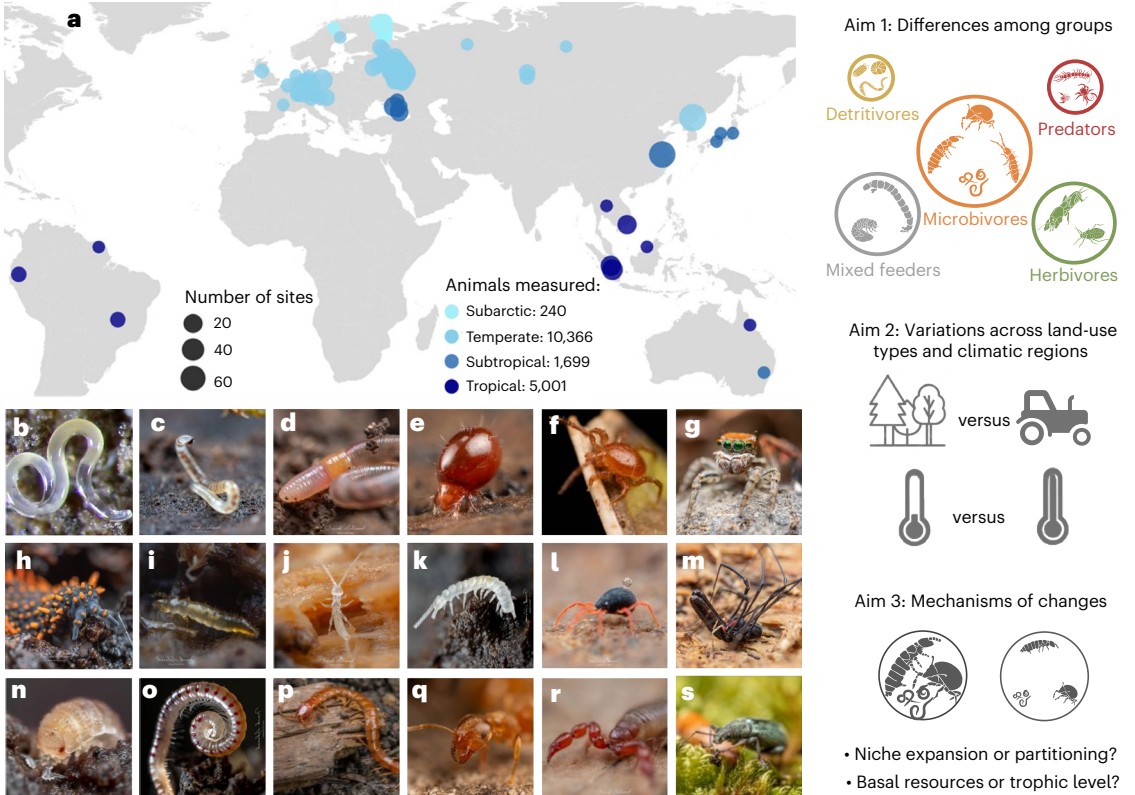

**Fig. 1 | Distribution of the 456 sampling sites across 19 countries. a**, Point size represents the number of sites at the respective locality, point colour represents the climatic zone. **b**–**s**, Representatives of the soil animals considered in this study, including nematodes (Nematoda) (**b**), enchytraeid worms (Enchytraeidae) (**c**), earthworms (Lumbricina) (**d**), moss mites (Oribatida) (**e**), predatory mites (Mesostigmata) (**f**), spiders (Araneae) (**g**), springtails (Collembola) (**h**), proturans (Protura) (**i**), diplurans (Diplura) (**j**), garden centipedes (Symphyla) (**k**), sucking mites (Prostigmata) (**l**), harvestmen (Opiliones) (**m**), woodlice (Isopoda) (**n**), millipedes (Diplopoda) (**o**), centipedes (Chilopoda) (**p**), ants (Formicidae) (**q**), false scorpions (Pseudoscorpiones) (**r**) and beetles (Coleoptera) (**s**). The right panel illustrates the hierarchical approach of the study: (1) assessing differences in trophic diversity among different functional groups of soil animals, (2) examining how trophic diversity of soil animals changes across biomes and land-use types and (3) understanding the mechanisms of changes in trophic diversity, for example, niche expansion or partitioning. In **a**, icons from Svenja Meyer and basemap data from Natural Earth (https://www.naturalearthdata.com). Photographs from Haifeng Yin (**b**) and Frank Ashwood (**c**–**s**).

the most accessible prey, which are often r-strategists characterized by high abundance, high metabolism and limited defence, such as springtails[57]. This similarity in prey selection probably contributes to the similarity of trophic niches among predators[58], which is also indicated by their smaller variations in $\delta^{13}C$ values (Supplementary Fig. 3a) and emphasizes their role in coupling different energy channels in soil food webs[6,59].

Herbivores had an intermediate trophic diversity and did not differ significantly from the other functional groups (Fig. 2a and Supplementary Table 1). They showed large variability in $\delta^{13}C$ but not in $\delta^{15}N$ values (Supplementary Fig. 3), indicating that the trophic diversity among belowground herbivores is predominantly a consequence of variations in the use of basal resources rather than trophic levels. Aboveground herbivore invertebrate taxa typically specialize in consuming specific plant species based on plant species-specific traits, including nutrient composition, N concentration and chemical defences, which can be attributed to coevolutionary dynamics between consumers and their host plants[60,61]. These food preferences based on plant species-specific traits might similarly apply to soil herbivores, which mainly feed on roots[62].

Overall, our results showed that trophic diversity and niche differentiation of soil animals depend on the position of functional groups within the food web. Functional groups that couple different energy channels, such as detritivores and predators, exhibit lower trophic diversity and niche differentiation.

## Higher trophic diversity in agricultural and tropical systems

Across the globe, intensive land use is considered a threat to soil biodiversity[26,40], with both the taxonomic and functional diversity declining with land-use intensity[39–41]. However, in contrast to our second hypothesis, the trophic diversity of soil animals tended to be greater in agricultural systems than in woodlands (on average by 32.1% ± 11.0%; Fig. 3a and Supplementary Table 6), being significantly greater by 36.3% ± 18.1%, 57.8% ± 16.7% and 63.2% ± 17.1% in detritivores, microbivores and predators, respectively. Previous studies using similar isotopic methods have shown that land use affects the trophic diversity of various animal groups, including fish, birds, mammals and zoobenthos, in diverse and context-dependent ways[32–36]. For example, trophic diversity in birds was found to be higher in disturbed (urban) than in natural ecosystems, as generalists exploited new niches created by human activities[32], which may also explain similar patterns in the present study. Agricultural land use typically reduces the supply of aboveground residues to soil animals as a result of the removal of crops, thereby aggravating resource shortage of soil animals[63,64]. However, agricultural land use is also associated with increased input of nutrients via fertilization, thereby potentially augmenting resource heterogeneity[29,65], which probably contributed to the larger variations in $\delta^{15}N$ than $\delta^{13}C$ values in agricultural than woodland ecosystems (Supplementary Fig. 4b). Note that because of the lack of detailed sampling dates in our dataset, the observed patterns in agricultural systems may reflect mixed conditions across different stages of the crop cycle,

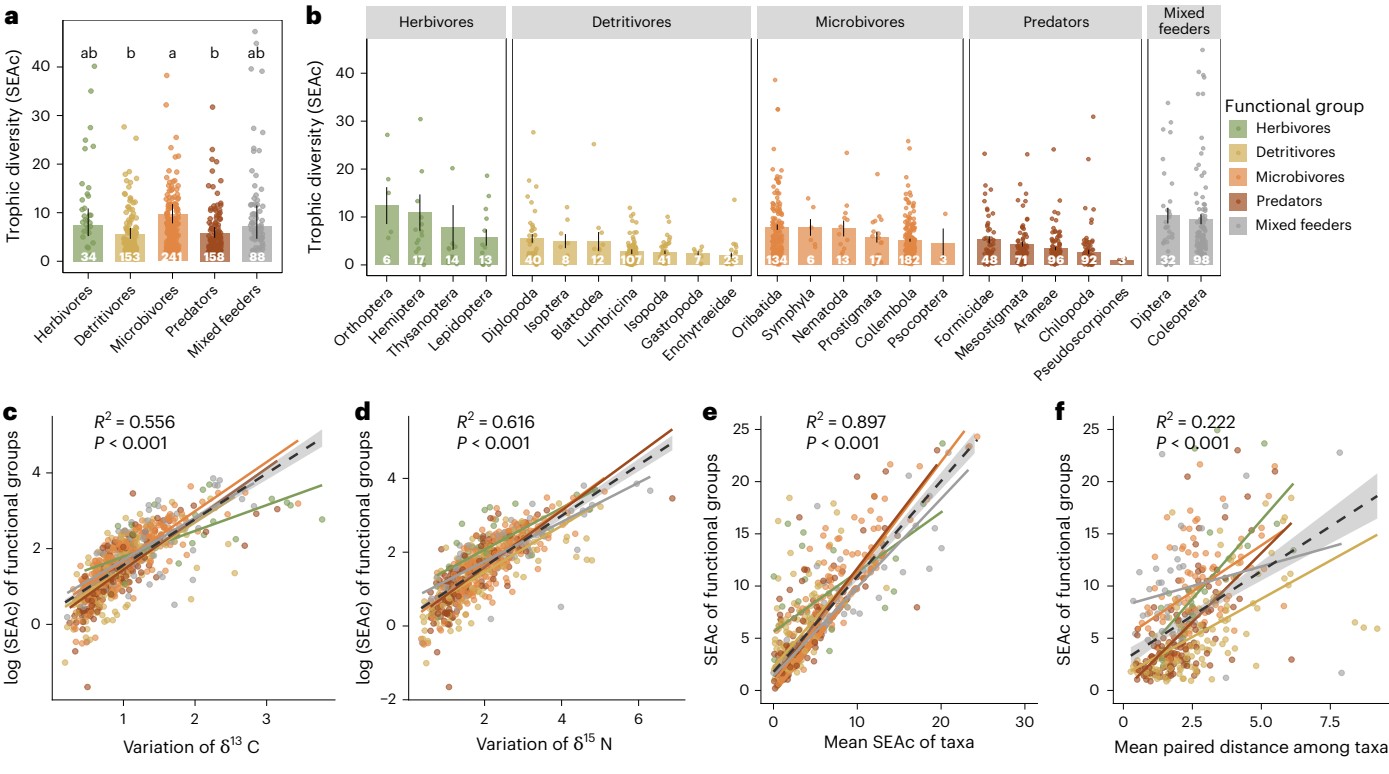

**Fig. 2 | Trophic diversity (corrected stable isotope ellipse area; SEAc) of different functional groups of soil animals (colour coded). a**, Trophic diversity of functional groups, shown as model-estimated means with 95% confidence intervals (CIs). **b**, Trophic diversity of each taxon, mean ± s.e., numbers in bars indicate the number of independent sampling sites, bars with different letters indicate significant differences. **c**, Relationship between log-transformed trophic diversity and variations in $\delta^{13}C$ values in each functional group. **d**, Relationship between log-transformed trophic diversity and variations in $\delta^{15}N$ values in each functional group. **e**, Relationship between trophic diversity of functional groups and the mean trophic diversity of taxa in each functional group. **f**, Relationship between trophic diversity of functional groups and the mean pairwise distance between the centroids of trophic positions of taxa in each functional group. Black lines denote overall model fit and coloured lines indicate different functional groups and shaded areas indicate 95% CIs of the fitted regression lines. Relationships were analysed using LMMs with two-sided tests. $R^2$ values represent the proportion of variance explained across functional groups. ***$P < 0.001$, exact $P$ values and full model results are provided in Supplementary Tables 1, 2, 4 and 5.

such as post-fertilization or post-harvest conditions. Although most soil animals are trophic generalists, they exhibit specific preferences for similar resources when resources are abundant, therefore being termed 'choosy generalists'[15,66,67]. Abundant resource supply might result in niche homogenization due to animals predominantly using the resources in ample supply as may be the case in woodlands, which typically have thicker litter layers compared with agricultural systems. Conversely, scarcity of resources may result in trophic differentiation by forcing animals to also exploit non-preferred resources[42]. In fact, agricultural land use has been shown to increase trophic diversity among individuals in springtail communities[68]. In agricultural systems, soil animals may partition their niches as a result of restricted and heterogeneous resource supply and may also opportunistically incorporate new resources[69], thus leading to higher trophic diversity at the community level. This was confirmed by lower trophic dissimilarity of the taxa within the same functional group (less niche partitioning, that is, niche homogeneity) in woodlands compared with agricultural systems (Supplementary Fig. 1b and Supplementary Table 3). Furthermore, the trophic dissimilarity between functional groups, such as microbivores and detritivores, was also larger in agricultural than in woodland ecosystems (Supplementary Fig. 5b). The potential mismatch between taxonomic diversity and trophic diversity indicates that soil animals may be able to expand their trophic niches under land-use changes (Supplementary Fig. 4a), thereby partly maintaining associated soil functions[11].

Supporting our second hypothesis, trophic diversity of soil animals tended to be greater in tropical than in temperate regions (on average by 40.6% ± 12.3%; Fig. 3b and Supplementary Table 6), being significantly greater by 61.4% ± 18.1%, 41.1% ± 16.6% and 68.6% ± 17.1% in detritivores, microbivores and predators, respectively. Recent global compilations reported soil animals, including macrofauna, mesofauna and microfauna, having lower density but higher taxonomic richness in the tropics than at higher latitudes[37–39]. Thus, the larger trophic diversity in the tropics may be related to increased taxonomic richness, which is also indicated by our results of increasing trophic diversity with taxon richness (Supplementary Fig. 6 and Supplementary Table 7). Further, it also aligns with higher trophic dissimilarity of taxa within the same functional group (niche partitioning) in the tropics compared with temperate systems (Supplementary Figs. 1b and 4a and Supplementary Table 3). However, even when accounting for the effect of taxonomic richness, effects of climate on trophic diversity remained strong (Supplementary Table 7). Presumably, at least in part this may be related to low accumulation of litter and soil organic matter in the tropics[70]. Low-latitude ecosystems such as tropical rainforests typically develop on old and weathered soils deficient in nutrients, being particularly phosphorus limited[71], which is also reflected by decreasing litter nutrient concentrations towards the tropics[72]. Similar to our study, others[49] reported that tropical riparian predators (predominantly spiders) had markedly higher trophic diversity than temperate ones, aligning with our findings and supporting the role

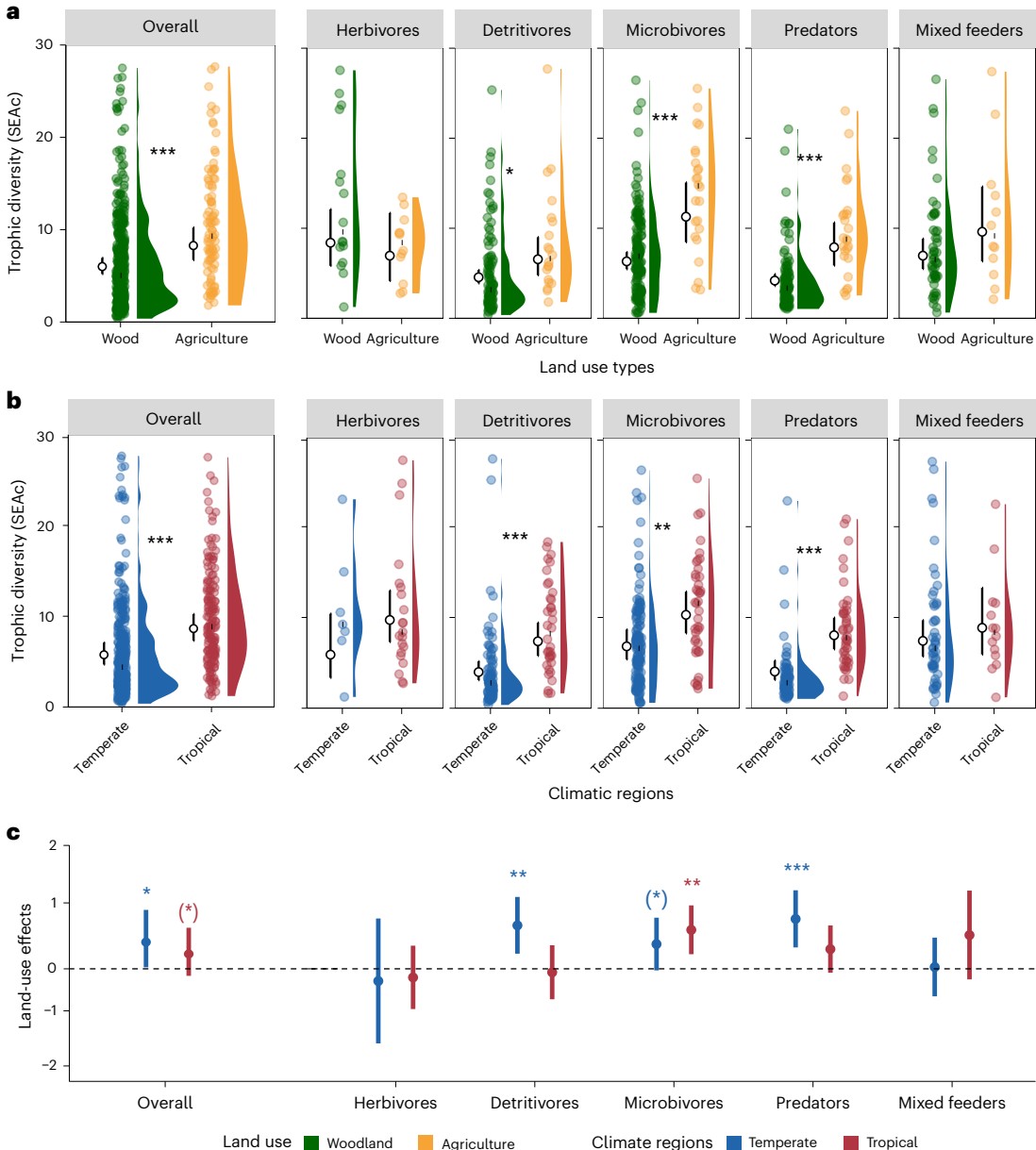

**Fig. 3 | Trophic diversity (SEAc) across functional groups and of each functional group of soil animals. a,b**, Different land-use types (woodlands and agricultural systems) (**a**) and climatic regions (temperate and tropical) (**b**), shown as model-estimated means with 95% CIs; points represent independent sampling replicate sites. **c**, Effects of land use on trophic diversity (SEAc) of different functional groups of soil animals in tropical and temperate regions; effect sizes are given as model-estimated means with log-response ratios (with 95% CIs) of contrasts between agriculture and woodland. Effects were analysed using LMMs with two-sided tests. (*)$P < 0.1$, *$P < 0.05$, **$P < 0.01$, ***$P < 0.001$, with exact $P$ values provided in Supplementary Table 6.

of niche expansion and partitioning in tropical ecosystems. Animals in the tropics exhibit higher metabolism and predation rates than those in high-latitude ecosystems, leading to intensified interactions and stress[39,73]. Consequently, generalist species may compete more intensely for high-quality food resources that are scarce. We also tested effects of land use and climate on trophic diversity at higher taxonomic resolution, namely at family, genus and species level. This analysis confirmed the pattern of higher trophic diversity in agricultural systems and in the tropics to be robust across taxonomic scales (Supplementary Fig. 7 and Supplementary Table 8). Thus, except for higher taxonomic richness, limitations in the quality and quantity of food resources and stronger competition for resources may drive niche partitioning among soil animals, as indicated by larger trophic dissimilarity at both the levels of high-ranking taxa and species. Therefore, the

higher trophic diversity of soil animals in the tropics is probably due to both niche partitioning and niche expansion.

In contrast to temperate systems, detritivores and predators showed no land-use effect on trophic diversity under resource-limited tropical conditions (Fig. 3c). This suggests that only under relatively high resource availability (temperate soils) these generalist feeders do exploit new niches under land uses, they partition resources as 'choosy generalists' and expand their trophic niche breadth under such conditions. By contrast, uniformly low resource quality and intense competition in tropical soils may generally constrain such niche differentiation irrespective of land-use system. Meanwhile, herbivores and mixed feeders exhibited minimal changes in trophic diversity across both land-use systems and climatic regions (no significant main or interactive effects; Fig. 3c and Supplementary Table 1). Herbivores

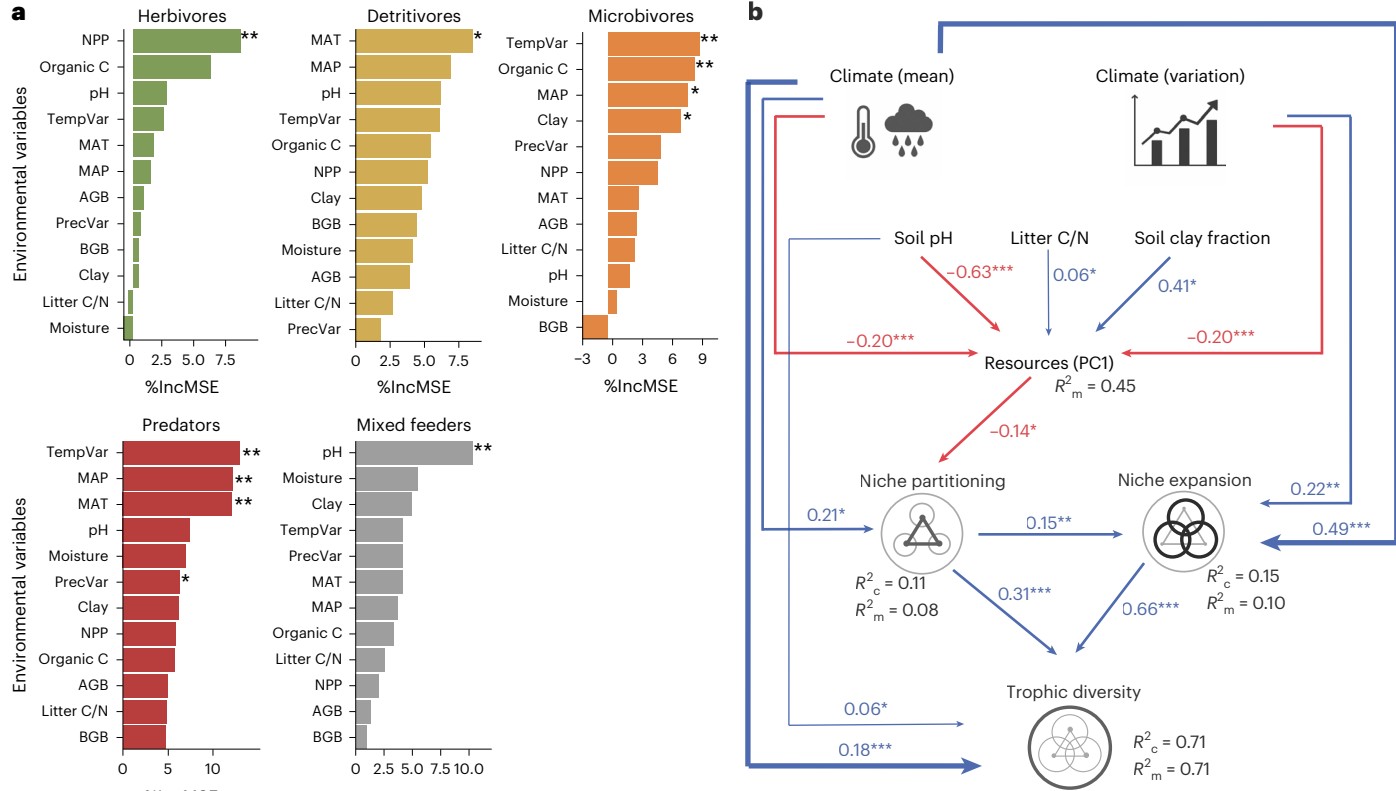

**Fig. 4 | Environmental drivers of trophic diversity among functional groups. a,b**, Random forest analysis (**a**) and the pathways showing how environmental factors affect the trophic diversity via niche partitioning (trophic dissimilarity among taxa) and niche expansion (trophic diversity of taxa) as indicated by piecewise structural equation models (piecewiseSEM) (**b**). Variable importance in random forest models was evaluated using permutation test, with exact $P$ values provided in Supplementary Table 9. In the piecewiseSEM, the conditional and marginal $R^2$ that is, $R_c^2$ and $R_m^2$, respectively, represent the proportion of variance explained by the variables without and with accounting for random effects of site. Red and blue colours of the arrows indicate negative and positive relationships, respectively; only significant relationships are shown. Numbers adjacent to arrows are standardized path coefficients. Significance for individual paths were obtained from two-sides Wald tests of fixed effects in the corresponding LMMs: *$P < 0.05$, **$P < 0.01$, ***$P < 0.001$, with exact $P$ values provided in Supplementary Table 10. The SEM adequately describes the data ($P = 0.395$, d.f. = 8, Fisher's $C = 8.40$). Clay, soil clay fraction (%); litter C/N, litter carbon-to-nitrogen ratio; moisture, water content in soil (%); organic C, soil organic carbon (gC soil kg$^{-1}$); PrecVar, precipitation variation of seasonality; TempVar, temperature variation of seasonality. Icons in **b** from Yan Zhang.

remain confined to plant-based energy channels, limiting their trophic flexibility and capacity for niche expansion, while the inherently broad diet (high trophic flexibility) of mixed feeders allows a consistent niche breadth regardless of environmental context.

### Environmental predictors of trophic diversity

Random forest analysis revealed that climatic factors, including temperature, precipitation and their seasonal variations, significantly influenced the trophic diversity of detritivores, microbivores and predators (Fig. 4a). This highlights the central role of climate in shaping the trophic diversity of soil animal communities. In addition, trophic diversity of microbivores was further related to resource availability (organic C) and soil structure (clay fraction), which probably reflects that trophic interactions between microbivores and microorganisms depend on soil organic matter as a substrate[15], with soil structure driving the accessibility of microbial prey by shaping soil pore architecture and moisture retention[50]. By contrast, the trophic diversity of herbivores was primarily explained by net primary productivity (NPP; gC m$^{-2}$)), probably because higher NPP supports a more diverse array of plant-derived resources for herbivory.

Structural equation modelling (SEM) supported that the increased trophic diversity of functional groups was due to climatic factors (climate principal component 1 (climate PC1), representing mean annual temperature (MAT) and mean annual precipitation (MAP) and

demonstrated that this was due to both enhanced niche partitioning and niche expansion (Fig. 4b and Supplementary Fig. 8). This aligns well with our observation that trophic diversity is higher in tropical compared with temperate regions. Further, increased resource availability (resources PC1, representing organic carbon and belowground biomass) negatively impacted trophic diversity by reducing niche partitioning, that is, abundant resources caused the trophic niches of different taxa to converge. Overall, the results highlight that climatic factors and resource availability shape trophic diversity, emphasizing their multifaceted influence on soil food webs.

### Conclusions and implications

On the basis of a large dataset on stable isotope ratios of soil animals, we analysed the trophic diversity of major soil animal functional groups and their variations across land-use systems and biomes. While informative, our findings should be interpreted in light of the limited geographic coverage of the dataset. We showed that microbivores are more trophically diverse than detritivores and predators, suggesting that the former play more diverse functional roles in soil food webs. Additionally, we showed that trophic diversity of soil animals is higher in agricultural systems than in woodlands despite the previously documented declines in biodiversity, suggesting that soil animals may broaden their trophic niches when facing resource shortages and frequent disturbances. The ability of soil animal communities

to broaden their trophic niches in response to global change, such as land use and climate change, may help to buffer ecosystems against instability by promoting resilience through diversified resource use. Specific soil animal functional groups, particularly microbial feeders, could enhance ecosystem functions such as nutrient cycling and decomposition by exploiting underused or rare resources. This flexibility in trophic niches suggests that soil communities may adapt in ways that maintain ecosystem functions. However, it also highlights potential risks, as increased flexible foraging behaviour may reflect the loss of specialists (ecological losers) from agricultural landscapes and their replacement by fewer generalists (ecological winners) being able to adapt to alternative resources, with potentially long-term implications for biodiversity and ecosystem functions[74–76].

## Methods

### Field sites and sampling

The study was based on extensive data collection and analysis across 456 study sites and 19 countries. About half the data were published before (55.7%)[53–55,68,77–99] and other data were compiled for this study. The dataset comprised 17,306 sample records of paired $\delta^{13}C$ and $\delta^{15}N$ values in soil animals distributed across four climatic regions: subarctic (240), temperate (10,366), subtropical (1,699) and tropical (5,001). The investigated ecosystem types included woodlands, agricultural systems and grasslands; the vegetation type and management details are provided in the animal_iso_core table (Data availability). The variations of SEAc of soil animals in different vegetation types are shown in Supplementary Fig. 9. Most of our dataset (95.0%) was generated by two collaborating research groups (primarily University of Göttingen and a close collaborator at the Institute of Ecology and Evolution, RAS, Moscow). Although we compiled a large dataset across multiple continents, the current dataset provides limited global representation. A substantial proportion of the data originate from Europe and tropical regions are comparatively under-represented relative to temperate regions. Certain regions, such as Africa and North America, remain under-sampled.

For details on the sampling methods for published data see refs. [53–55,68,77–99]. For unpublished data, standard extraction methods were used. Nematodes were sampled by extracting 5-cm diameter soil cores encompassing the litter layer and the top 0–5 cm of the mineral soil and were extracted by using wet extraction with Baermann funnels. Soil mesofauna and macrofauna were sampled by using heat Berlese or Kempson extractors[100] and preserved in 70–96% ethanol. Sampling methods deviations are listed in the animal_iso_core table (Data availability).

Animals were classified into 26 high-rank taxonomic groups and further into five major functional groups as follows: herbivores (Hemiptera, Orthoptera, Thysanoptera and Lepidoptera), detritivores (Lumbricina, Diplopoda, Isopoda, Isoptera, Dermaptera, Blattodea, Gastropoda and Enchytraeidae), microbivores (Collembola, Oribatida, Nematoda, Protura, Prostigmata, Psocoptera and Symphyla) and predators (Araneae, Chilopoda, Diplura, Formicidae, Mesostigmata, Opiliones and Pseudoscorpiones) and groups showing mixed feeding (Diptera and Coleoptera)[15,101]. It has been shown that high-rank animal taxa in soil typically are trophically and functionally consistent[101].

We collected a suite of environmental covariates for each of our 456 sampling locations. These covariates included climate, soil physicochemical properties, vegetation productivity and litter quality indices. Litter carbon-to-nitrogen (C:N) ratios were calculated from laboratory measurements of total carbon and total nitrogen content in the local litter. Climatic variables, including MAT (°C), MAP (mm), temperature seasonality and precipitation seasonality, were obtained from the WorldClim v.2 bioclimatic dataset at 30-arcs (~1 km) resolution. The gradients of MAT and MAP of the study sites were shown in Supplementary Fig. 10. These variables were accessed and extracted using Google Earth Engine (GEE): soil variables were derived from OpenLandMap and included topsoil clay fraction, pH (in $H_2O$), organic carbon (g kg$^{-1}$), volumetric water content at 33 kPa and US Department of Agriculture texture class; vegetation productivity was characterized using annual NPP (gC m$^{-2}$ yr$^{-1}$) from the MODIS/Terra MOD17A3HGF product (500-m resolution, Collection 006), with values averaged or selected from the corresponding sampling year; aboveground (AGB) and belowground (BGB) biomass carbon densities (originally in MgC ha$^{-1}$) were extracted from the NASA ORNL biomass carbon density dataset and converted to kgC m$^{-2}$; all spatial layers were reprojected to a common geographic coordinate system (WGS 84, EPSG:4326) and sampled using the reduceRegions() function in GEE, using a spatial resolution of 250 m and nearest-neighbour resampling.

### Stable isotope analysis

Animals were identified to family-level (86.9%), genus-level (70.1%) or species-level (58.5%) before being processed for stable isotope analysis. Before stable isotope analysis, animals were dried at 50–60 °C for 24 h, then weighed and enclosed in tin capsules; sample weights ranged from 0.01 mg to 1.0 mg. For small-sized animals, the whole body of individual animals were used for stable isotope analysis, with several individuals bulked when more biomass was required, for large-sized animals we used body parts dominated by muscle tissue (for example, legs)[102]. Stable isotope ratios of $^{13}C/^{12}C$ and $^{15}N/^{14}N$ were measured using a system comprising an elemental analyser and a mass spectrometer. Ratios between the heavy isotope and the light isotope ($^{13}C/^{12}C$, $^{15}N/^{14}N$; R) were presented in parts per thousand relative to the standard using the delta notation, denoted as $\delta^{13}C$ or $\delta^{15}N = (R_{sample}/R_{standard} - 1) \times 1000$ (‰). Vienna PD Belemnite and atmospheric nitrogen served as the standards for $^{13}C$ and $^{15}N$, respectively. Isotope measurements were calibrated using international reference materials (IAEA-600 caffeine, IAEA-CH-6 sucrose, IAEA-N2 ammonium sulfate, USGS-40 glutamic acid), with analytical precision for $\delta^{15}N$ approximately ±0.2‰ (s.d.) across all runs. Accuracy was ensured as measured $\delta^{15}N$ values typically differed ≤0.2‰ from certified standards, ensuring minimal bias even for small-mass samples. In case of small sample mass, our isotope-ratio mass spectrometry setup was specifically optimized for low nitrogen content samples, using micro tin capsules, extended combustion times and reduced blank signals, as detailed in ref. [103]. We ensured measurement quality by retaining only samples with the ratio between measurements of the mass spectrometer and the thermal conductivity detector of the gas chromatograph between 0.99 and 1.01. All reported $\delta^{13}C$ and $\delta^{15}N$ values are already baseline-corrected against international standards.

### Calculation of trophic diversity and dissimilarity

Trophic diversity of soil animals can be determined by computing the standard ellipse area (SEA) on the basis of position of soil animals within the $\delta^{13}C$–$\delta^{15}N$ biplot of taxonomic and functional groups at each site, using a Bayesian framework implemented in the SIBER package in R. We used corrected standard ellipse area (SEAc) instead of SEA in our study, which is more robust in handling small and variable sample sizes than SEA[45]. The relationship between SEA and SEAc can be formulated as SEAc = SEA$(n_{sample size} - 1)(n_{sample size} - 2)^{-1}$. We visualize some examples of the SEAc of detritivores, microbivores and predators in woodland and the agricultural systems by randomly picked five sites for each group (Supplementary Fig. 11). Moreover, to further limit potential bias stemming from small sample sizes, ellipses were exclusively computed for taxonomic and functional groups that consisted of five or more samples per site. To assess trophic dissimilarity among taxonomic groups within each functional group, we calculated the mean pairwise distance between the centroids of isotopic positions of taxonomic groups within each functional group[46]. Taxonomic-level SEAc and between-taxon trophic dissimilarity were used as indicators of niche expansion and niche partitioning, respectively, in the SEM and

other analyses to identify the mechanisms driving changes in trophic diversity at the functional group level[32].

We used uncalibrated stable isotope values ($\delta^{13}$C and $\delta^{15}$N) for assessing trophic diversity and trophic dissimilarity of soil animals, as calibration using litter $\delta^{13}$C and $\delta^{15}$N values did not significantly affect SEAc and trophic dissimilarity. We calculated the standard deviation of $\delta^{13}$C and $\delta^{15}$N values within the same functional group at each site. These values served as indicators of the variation in both basal resource use and trophic position among functional groups.

## Statistical analyses

All analyses were done in R 4.0.3[104]. To assess the effects of land use and climate on the SEAc and trophic dissimilarity of soil animals, we selected the subsets from tropical/temperate and agricultural systems/woodlands from the whole dataset. These two climatic zones and land-use types were selected because they had the most robust sample sizes; however, in the subsequent analyses of environmental drivers of trophic diversity (SEM and random forest) we used the full dataset. We fitted linear mixed-effects models (LMMs) using log-transformed SEAc and trophic dissimilarity and then applied contrasts between tropical and temperate ecosystems, as well as between agricultural systems and woodlands to estimate effect sizes. We conducted three separate LMMs for log-transformed SEAc of functional groups, SEAc of taxonomic groups and trophic dissimilarity. The models included functional groups (herbivores, detritivores, microbivores and predators), land use (agricultural systems and woodlands), biome (tropical and temperate) and their interactions as fixed effects. To account for non-independence, we included the study site identity (siteID) as a random effect in all models to account for the non-independence of multiple observations (across different functional groups) coming from the same location. The siteID refers to the unique identifier of the sampling site and is also the spatial unit at which the trophic diversity of soil animals was calculated. Spatial autocorrelation was tested by using Moran's $I$ on model residuals based on the five nearest neighbours of sampling coordinates, which indicated no significant spatial autocorrelation ($P = 0.18$); thus, spatial autocorrelation was not included in the final models. In addition, we included taxonomic group identity (nested within functional group) as random effect in models using taxonomic group-level SEAc and taxonomic group identity (reflecting the taxonomic composition of each functional group) as random effect in models using functional group-level SEAc.

For estimating effect sizes of land use and climate, we used the emmeans package to compute the estimated marginal means in the linear models. Then, we used the contrast function from the emmeans package to calculate contrasts between temperate versus tropical and woodland versus agriculture[105].

Additionally, we built another model and included sampling number and family richness as covariates to inspect their effects on the log-transformed SEAc of functional groups. The models included functional groups, land use, climate, sampling number and family richness as fixed effects, with site included as random effect. We checked model assumptions of the most parsimonious models by fitting model residuals versus the results of fitted models.

To elucidate the drivers behind larger functional group SEAc, we used two separate LMMs. One model included SEAc of taxonomic groups and trophic dissimilarity as explanatory variables, while the other explored variations in $\delta^{13}$C and $\delta^{15}$N as separate explanatory variables. We additionally evaluated models incorporating squared $\delta^{13}$C and $\delta^{15}$N terms, which showed inferior fit (higher Akaike information criterion/Bayesian information criterion and lower log-likelihood) compared with the linear specifications, thus justifying our use of linear terms. We then used the estimated value of the coefficient for each independent variable to estimate their contribution to SEAc of functional groups.

To estimate the environmental drivers of trophic diversity of soil animals, random forest models were implemented to build a set of regression trees with environmental predictors and to average the results[106] for each functional group. In total, 12 environmental predictors were used to assess their explanatory power for explaining the trophic diversity with percentages of increased mean stand error (%IncMSE). The number of the trees was set to 500 and the minimum node size was set to three.

For further exploring the pathways of environmental variables working on trophic diversity via niche expansion (trophic diversity of taxa) and niche partitioning (trophic dissimilarity among taxa), piecewise structural equation models (piecewiseSEM) were used. We mapped the hypothesized causal pathways from climatic/edaphic factors directly to the niche partitioning/expansion or via resources (Supplementary Fig. 12). To comprehensively indicate variables of climate, climate variation and resources, we performed principal component analysis for these variables, respectively (Supplementary Fig. 12). We combined MAP and MAT with PC1 of them for climate (mean) and combined temperature variation of seasonality and precipitation variation of seasonality with PC1 of them for climate variation and combined belowground biomass, soil organic C and net primary productivity, with PC1 of them for resources. With larger PC1 values of climate (mean), climate variation and resources mean higher temperature and precipitation, larger climatic variation and more resources, respectively (Supplementary Fig. 13). We used piecewiseSEM rather than standard SEM because it allowed us to consider the random factor of site[107]. On the basis of our priori model, we modified models based on hypothesized causal pathways (Supplementary Fig. 12) until the directional separation tests and Fisher's $C$ test passed ($P > 0.05$). Then, we removed the paths that caused problems of negative Chi-square, poor causal links (directional separation tests, $P < 0.05$) and poor model fit (Fisher's $C$ test, $P < 0.05$) to remain consistent with the hypothesized model as much as possible (Supplementary Fig. 12).

The SIBER package was used for calculating the trophic diversity of soil animals[45]. The lme4 package was used to fit LMMs[108] and the emmeans package to compute the estimated marginal means in the linear models[105]. The randomForest[106] package was used to conduct the random forest models. The packages nlme, lme4 and piecewiseSEM[109] were used to fit piecewiseSEM. All mixed models were visually checked to meet the assumption of residual homogeneity of variance. Results were visualized using the ggplot2 package[110].

### Reporting summary

Further information on research design is available in the Nature Portfolio Reporting Summary linked to this article.

## Data availability

The data used in this work are available via figshare at https://figshare.com/s/c4a378183d4d35e982d1 (ref. 111).

## Code availability

The R code used in this work is available via figshare at https://figshare.com/s/c4a378183d4d35e982d1 (ref. 111).

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

## Acknowledgements

The work was supported by the Alexander von Humboldt foundation in the framework of a Research group linkage programme 1071297-RUS-IP 'Structure and functioning of belowground food webs across temperate and tropical eco- systems'. Z.Z. was supported by Deutsche Forschungsgemeinschaft (DFG, 532858005) and the China Scholarship Council (CSC, 202004910314). A.P. was supported by the DFG, German Research Foundation, in the framework of the Emmy Noether programme (project no. 493345801) and iDiv (DFG–FZT118, 202548816). S.S., A.D.B. and M.J. acknowledge support by the DFG in the framework of the collaborative German–Indonesian research project CRC990—EFForTS (192626868—SFB 990). S.S. and A.P. further acknowledge funding by DFG project 532846413. M.M.P. was funded by the DFG Priority Program 1374 'BiodiversityExploratories' (SCHE 376/38-2). N.E. acknowledges funding by the DFG (German Centre for Integrative Biodiversity Research, FZT118; Ei 862/29-1; Ei 862/31-1). K.S. acknowledges CSF 22-17593M and ERC 805189. D.K. was supported by RSF (25-24-00639). We thank F. Ashwood and H. Yin for the soil animal photos and S. Meyer for the animal silhouettes.

## Author contributions

Z.Z. and A.P. conceptualized the idea and designed the study. Z.Z., A.D.B., M.M.P., M.J., Y.Z., U.B., F.H., N.S., O.S., Y.H., B.K., A.A.G., A.K., D.K., A.G., I.I.L., S.M.T., J.S., M.S., V.A.Z., O.L.R., W.I.S., F.V.C., D.A.P., Z.L., C.M., Z.X., D.W., M.M., K.S., A.V.T., S.S. and A.P. collected data. A.P., S.M.T. and Z.Z. compiled the data. Z.Z. conducted the data analysis with help from A.P., Y.Z. and N.E. Z.Z. wrote the paper. All authors contributed to interpretation of the results and edited drafts of the paper.

## Funding

## Competing interests

The authors declare no competing interests.

## Additional information

**Correspondence and requests for materials** should be addressed to Zheng Zhou.

[1]Animal Ecology, University of Göttingen, Göttingen, Germany. [2]Ecology of Tropical Agricultural Systems, University of Hohenheim, Stuttgart, Germany. [3]German Centre for Integrative Biodiversity Research (iDiv), Leipzig, Germany. [4]Institute of Biology, Leipzig University, Leipzig, Germany. [5]Te Aka Mātuatua - School of Science, University of Waikato, Hamilton, New Zealand. [6]Julius Kühn Institute, Federal Research Centre for Cultivated Plants, Institute for Forest Protection, Quedlinburg, Germany. [7]Department of Global Change Ecology, Biocenter, University of Würzburg, Würzburg, Germany. [8]Center for

Biodiversity and Integrative Taxonomy (KomBioTa), University of Hohenheim, Stuttgart, Germany. [9]Institute of Biodiversity, Friedrich-Schiller-University Jena, Jena, Germany. [10]Faculty of Environmental, Life, Natural Science and Technology, Okayama University, Okayama, Japan. [11]Senckenberg Museum of Natural History Görlitz, Görlitz, Germany. [12]UCD School of Agriculture and Food Science Centre, University College Dublin, Dublin, Ireland. [13]A.N. Severtsov Institute of Ecology and Evolution, Russian Academy of Sciences, Moscow, Russia. [14]UCD School of Biology and Environmental Science, University College Dublin, Dublin, Ireland. [15]Institute of Systematics and Ecology of Animals of Siberian Branch of Russian Academy of Sciences (ISEA SB RAS), Novosibirsk, Russia. [16]Department of Natural Sciences, Novosibirsk State University, Novosibirsk, Russia. [17]Institute for Alpine Environment, Eurac Research, Bozen, Italy. [18]Universität Innsbruck, Department of Ecology, Innsbruck, Austria. [19]Lobachevsky State University, Nizhny Novgorod, Russia. [20]Global Land Team, Ricardo Energy and Environment, Didcot, UK. [21]Key Laboratory of Urban Environment and Health, Institute of Urban Environment, Chinese Academy of Sciences, Xiamen, China. [22]Key Laboratory of Vegetation Ecology, Ministry of Education, Northeast Normal University, Changchun, China. [23]Biology Centre, Czech Academy of Sciences, Institute of Entomology, České Budějovice, Czech Republic. [24]Faculty of Science, University of South Bohemia, České Budějovice, Czech Republic. [25]Southern Branch, Joint Russian-Vietnamese Tropical Science and Technology Research Center, Ho Chi Minh City, Vietnam. [26]Centre of Biodiversity and Sustainable Land Use, University of Göttingen, Göttingen, Germany. [27]International Institute Zittau, TUD Dresden University of Technology, Dresden, Germany. [28]These authors contributed equally: Alexei V. Tiunov, Stefan Scheu, Anton Potapov. ✉e-mail: zzhou@gwdg.de

# Reporting Summary

## Statistics

For all statistical analyses, confirm that the following items are present in the figure legend, table legend, main text, or Methods section.

| n/a | Confirmed | |
|---|---|---|
| ☐ | ☒ | The exact sample size (*n*) for each experimental group/condition, given as a discrete number and unit of measurement |
| ☐ | ☒ | A statement on whether measurements were taken from distinct samples or whether the same sample was measured repeatedly |
| ☐ | ☒ | The statistical test(s) used AND whether they are one- or two-sided<br>*Only common tests should be described solely by name; describe more complex techniques in the Methods section.* |
| ☐ | ☒ | A description of all covariates tested |
| ☐ | ☒ | A description of any assumptions or corrections, such as tests of normality and adjustment for multiple comparisons |
| ☐ | ☒ | A full description of the statistical parameters including central tendency (e.g. means) or other basic estimates (e.g. regression coefficient) AND variation (e.g. standard deviation) or associated estimates of uncertainty (e.g. confidence intervals) |
| ☐ | ☒ | For null hypothesis testing, the test statistic (e.g. *F*, *t*, *r*) with confidence intervals, effect sizes, degrees of freedom and *P* value noted<br>*Give P values as exact values whenever suitable.* |
| ☒ | ☐ | For Bayesian analysis, information on the choice of priors and Markov chain Monte Carlo settings |
| ☒ | ☐ | For hierarchical and complex designs, identification of the appropriate level for tests and full reporting of outcomes |
| ☒ | ☐ | Estimates of effect sizes (e.g. Cohen's *d*, Pearson's *r*), indicating how they were calculated |

*Our web collection on statistics for biologists contains articles on many of the points above.*

## Software and code

Policy information about availability of computer code

| Data collection | no software was used for collecting the data |
|---|---|
| Data analysis | Analysis was implemented in R v4.2.0 with R studio interface v1.4.1103 (RStudio, PBC). The following packages were used: SIBER v2.1.9, ggpubr v0.6.0, maps v3.4.1, vegan v2.6-8, ggplot2 v3.5.1, nlme v3.1-162, effects v4.2-2, dplyr v1.1.4, tidyr v1.3.0, readxl v1.4.2, lme4 v1.1-32, lmerTest v3.1-3, emmeans v1.10.4. Linear models are specified in the Extended Data Table 1-5, 7 and 8. The code for figures and statistics are available here in figshare: https://figshare.com/s/c4a378183d4d35e982d1 |

For manuscripts utilizing custom algorithms or software that are central to the research but not yet described in published literature, software must be made available to editors and reviewers. We strongly encourage code deposition in a community repository (e.g. GitHub). See the Nature Portfolio guidelines for submitting code & software for further information.

## Data

Policy information about availability of data

All manuscripts must include a data availability statement. This statement should provide the following information, where applicable:
- Accession codes, unique identifiers, or web links for publicly available datasets
- A description of any restrictions on data availability
- For clinical datasets or third party data, please ensure that the statement adheres to our policy

Data are avaiable here: https://figshare.com/s/c4a378183d4d35e982d1

## Research involving human participants, their data, or biological material

Policy information about studies with human participants or human data. See also policy information about sex, gender (identity/presentation), and sexual orientation and race, ethnicity and racism.

| | |
|---|---|
| Reporting on sex and gender | N/A |
| Reporting on race, ethnicity, or other socially relevant groupings | N/A |
| Population characteristics | N/A |
| Recruitment | N/A |
| Ethics oversight | N/A |

Note that full information on the approval of the study protocol must also be provided in the manuscript.

# Field-specific reporting

Please select the one below that is the best fit for your research. If you are not sure, read the appropriate sections before making your selection.

☐ Life sciences      ☐ Behavioural & social sciences      ☒ Ecological, evolutionary & environmental sciences

For a reference copy of the document with all sections, see nature.com/documents/nr-reporting-summary-flat.pdf

# Ecological, evolutionary & environmental sciences study design

All studies must disclose on these points even when the disclosure is negative.

| | |
|---|---|
| Study description | The study explores the trophic diversity of soil animals between different functional groups, land-use and climatic systems. |
| Research sample | The study analyses 26 high-rank taxonomic groups of soil animals (including earthworms, nematodes, spiders, etc) |
| Sampling strategy | The dataset comprised 15,893 sample records of paired δ13C and δ15N values in soil animals across 343 study sites and 15 countries, The investigated ecosystem types included woodlands, agricultural systems and grasslands. |
| Data collection | Authors of the paper collected the soil animals and conducted the stable isotope analysis. |
| Timing and spatial scale | Samples distributed across four climatic regions: subarctic, temperate, subtropical, and tropical regions, and from 2002 - 2018. |
| Data exclusions | No data were excluded from the analysis |
| Reproducibility | R code and statistical model specifications are openly available allowing to reproduce the data analysis: https://figshare.com/s/c4a378183d4d35e982d1 |
| Randomization | In this study, randomization was not applicable as the data were derived from a combination of published datasets and field sampling. Data collection followed standardized methods, with soil animals sampled systematically from the litter layer and topsoil using established protocols, ensuring consistency across locations. Further details of the sampling methods and sites are provided in Table S8. |
| Blinding | Blinding was not relevant to this study as it involved ecological data collected across multiple geographic locations and climatic regions. Data analysis was conducted objectively using numerical values (e.g., stable isotope ratios, climatic variables) without subjective interpretation that would necessitate blinding. |

Did the study involve field work?      ☒ Yes      ☐ No

## Field work, collection and transport

| | |
|---|---|
| Field conditions | Fieldwork was conducted across 343 sites spanning subarctic, temperate, subtropical, and tropical regions. Key abiotic parameters, including mean annual precipitation and temperature, were recorded using WorldClim data based on site coordinates. The details of study sites are listed in Table S8. |
| Location | Sampling locations were distributed across 15 countries, with precise latitude and longitude details for each site listed in Table S8. |

| Access & import/export | All sampling was conducted in compliance with local, national, and international regulations. Necessary permits for sampling, transportation, and export of samples were obtained where required. |
|---|---|
| Disturbance | Work on the study sites was implemented with care, to minimize disturbance. Whenever possible, manipulations with samples were done in a laboratory, outside the field sampling areas. |

# Reporting for specific materials, systems and methods

We require information from authors about some types of materials, experimental systems and methods used in many studies. Here, indicate whether each material, system or method listed is relevant to your study. If you are not sure if a list item applies to your research, read the appropriate section before selecting a response.

## Materials & experimental systems

| n/a | Involved in the study |
|---|---|
| ☒ | ☐ Antibodies |
| ☒ | ☐ Eukaryotic cell lines |
| ☒ | ☐ Palaeontology and archaeology |
| ☐ | ☒ Animals and other organisms |
| ☒ | ☐ Clinical data |
| ☒ | ☐ Dual use research of concern |
| ☒ | ☐ Plants |

## Methods

| n/a | Involved in the study |
|---|---|
| ☒ | ☐ ChIP-seq |
| ☒ | ☐ Flow cytometry |
| ☒ | ☐ MRI-based neuroimaging |

## Animals and other research organisms

Policy information about studies involving animals; ARRIVE guidelines recommended for reporting animal research, and Sex and Gender in Research

| Laboratory animals | Study did not involve laboratory animals |
|---|---|
| Wild animals | Only soil invertebrate animals (arthropods and earthworms) were collected and killed using ethanol during the study. This was necessary to assess stable isotope composition. We collected soil arthropod and earthworm communities using Kempson extractors. |
| Reporting on sex | Sex was not considered in the study |
| Field-collected samples | Collected soil samples were transported in the lab for heat extraction. No field-collected environmental samples were used in this study. |
| Ethics oversight | No ethical approval was required. The study did not involve vertebrate animal capturing and killing. |

Note that full information on the approval of the study protocol must also be provided in the manuscript.

## Plants

| Seed stocks | N/A |
|---|---|
| Novel plant genotypes | N/A |
| Authentication | N/A |

