## [Peer Review File · Nature Ecology & Evolution]

Greater trophic diversity of soil animal communities under agricultural land use and tropical climate

Corresponding Author: Dr Zheng Zhou

This manuscript has been previously reviewed at another journal. This document only contains information relating to versions considered at Nature Ecology & Evolution.

This manuscript has been previously reviewed at another journal that is not operating a transparent peer review scheme. The manuscript was considered suitable for publication without further review at Nature Ecology & Evolution.

Version 0:

Decision Letter:

8th December 2025

Dear Zheng,

Thank you for submitting your revised manuscript entitled "Greater trophic diversity of soil animal communities under agricultural land use and warmer climate" (NATECOLEVOL-25124307-T). Now that I have had a chance to examine it and the reviewers' comments again, I am pleased to confirm that we'll be happy in principle to publish it in Nature Ecology & Evolution, pending minor revisions to satisfy the reviewers' final requests and to comply with our editorial and formatting guidelines.

[redacted]

Referee #2 (Remarks to the Author):

I enjoyed the earlier version of this paper by Zhou et al. and thought the focus on soil animals and the scale of the work was impressive. Like the other reviewers though, I was concerned about the geographical bias in data and the superficial nature of the analysis of drivers. I also found the finding that trophic diversity is higher for microbivores than detritivores and predators to be unsurprising in view of past work and questioned the very general nature of the land use categories used.

The authors have gone to much effort to address these and the points of the other reviewers, and the paper is strengthened from this - the inclusion of more sites from underrepresented areas has improved the balance of the dataset, even though global representation is still limited as acknowledged; the presentation of data on trophic diversity across vegetation types provides some additional information on variation within the broad categories; the statistical analysis has been expanded to support conclusions on how different functional groups are influenced by a wider range of environmental variables and the role of resource supply for trophic diversity; and, last, the interpretation of data related to niche expansion and partitioning is better explained.

The revisions have much improved the paper, which now makes a valuable contribution as the first large scale assessment of soil animal trophic diversity. But I am still left questioning if the findings are of outstanding scientific importance as needed for publication in Nature. The conclusion that trophic diversity is higher for microbial than for detritus feeders and predators is interesting, but to be expected. The finding that trophic diversity is greater in tropical compared to temperate climates is also to be expected, given past work showing that taxonomic diversity of soil animals is higher in the tropics. As raised in my original review, it is unclear what can be learned from the broad finding that trophic diversity increases in agricultural ecosystems compared to woodlands which is so general given the likely big influence of the type and intensity of agricultural interventions (I couldn't find any management details in Table S9). Last, conclusions focus on implications for topics like resilience and adaptation, which is very speculative as not studied.

This study is valuable, especially with its scale, and the revision is much improved and will no doubt be well cited when published. I am less sure though about whether the conclusions are of high enough scientific importance to merit publication in Nature.

Response: We thank Referee #2 for recognizing the importance, scale, and novelty of the study and note that no technical or methodological concerns were raised.

We briefly respond to the referee's comments regarding the perceived novelty of the findings:

1. With respect to the higher trophic diversity of microbivores, this pattern has often been assumed based on intuition, but has not previously been demonstrated empirically across trophic groups. Our study provides the first cross-group, global-scale evidence supporting this assumption.
2. Similarly, higher trophic diversity in tropical compared to temperate regions should not be considered self-evident based solely on higher taxonomic diversity. Taxonomic and trophic diversity are not necessarily coupled, as illustrated by agricultural ecosystems, which often show reduced taxonomic diversity yet, in our analysis, exhibit relatively high trophic diversity.
3. Regarding agricultural systems, we incorporated a broad range of environmental variables and provide mechanistic interpretations of how resource supply and environmental conditions may shape trophic diversity, while acknowledging that management-specific effects could not be resolved at the global scale.

Overall, while some broad patterns may align with intuitive expectations or prior assumptions, this does not diminish their novelty or scientific importance, as scientific understanding cannot rely on intuition alone. Furthermore, the central contribution of this study lies in providing the first global, functionally explicit quantification of soil animal trophic diversity across multiple trophic groups within a unified analytical framework, enabling direct cross-group comparisons that have not previously been demonstrated empirically.

Referee #3 (Remarks to the Author):

The authors have done a nice job of responding to reviews, and I think that this can be a strong contribution to the literature. Additional suggestions for the authors' consideration:

Response: We thank Referee #3 for the supportive and constructive comments on the manuscript. The comments are addressed point by point below.

Title and Abstract: I still believe that emphasizing "under warmer climate" in your title and abstract (line 60) is misleading. The climatic PCA axis 1 was composed not only of the mean temperature, but also of precipitation, temperature variability etc. In fact, the mean annual temperature was significant only for detritivores and predators. More appropriate would be to discuss "under tropical climate" (or different climatic regions) as you do in the rest of the manuscript.

Response: Thank you for this suggestion. We have revised the title and abstract accordingly.

Line 98: I would suggest deleting: "trait-based functional" phrase, to streamline the sentence.

Response: Done.

Line 113: I would remove: "(niche expansion and partitioning)" from here, as it is elaborated on in the last sentence of this paragraph.

Response: Done.

Line 114: I would also remove "in soil food webs" as this applied more generally.

Response: Thanks. Done.

Figure 1 – as you hypothesised that herbivores (and microbiovores) have greater trophic diversity, I would make the herbivore ellipse in the Aim 1 larger (to visually highlight this).

Response: Thanks. Done as suggested.

Lines 125-129: Are you discussing "biomes" or "climatic regions" (see line 122 and Fig. 3). I would suggest to unite your terminology.

Response: We use the term "biomes" to refer broadly to ecosystem types that integrate both climatic conditions and land-use categories, rather than climatic regions alone.

Line 161: delete "occupation"

Response: Done.

Line 163: I would delete the first "trophic" on this line, changing to "are analogues of animals..."

Response: Done.

Fig. 2: I would indicate (at least in the panel a) which groups are significantly different from each other.

Response: Done as suggested.

Line 259: Change to "(predominantly spiders)"

Response: Done.

Line 358: Change "were showed" to "are shown"

Response: Done.